# Effects of Aging Uncertainty on the Estimation of Growth Functions of Major Tuna Species

**Dongqi Lu [1], Qinqin Lin [1], Jiangfeng Zhu [1,2,3,*] and Fan Zhang [1,*]**

[1]  College of Marine Sciences, Shanghai Ocean University, Shanghai 201306, China
[2]  Key Laboratory of Sustainable Exploitation of Oceanic Fisheries Resources, Ministry of Education, Shanghai 201306, China
[3]  Key Laboratory of Oceanic Fisheries Exploration, Ministry of Agriculture and Rural Affairs, Shanghai 201306, China
*   Correspondence: jfzhu@shou.edu.cn (J.Z.); f-zhang@shou.edu.cn (F.Z.)

**Abstract:** Fishery stock assessment requires accurate specification of the growth function of target species, and aging uncertainty is an important factor that affects the estimation of growth parameters. In this study, we used simulations to study the effects of two types of aging uncertainty, aging error and sampled age range, on the parameter estimation of the Von Bertalanffy growth function, including asymptotic length ($L_\infty$), growth coefficient ($k$), and theoretical age in the year at zero length ($t_0$) of five important tuna species. We found that the uncertainty of the estimated growth curves increased with increasing aging errors. When aging errors were fixed among ages, the effects of age range on estimation error of growth parameters were different among species and growth parameters. When the aging error increased with age, the estimation uncertainty of $L_\infty$ and $k$ was the greatest when only young age groups were sampled, while the estimation uncertainty of $t_0$ was the greatest when only old age groups were sampled. Therefore, reducing the aging error and sampling individuals with a wider age range are important for increasing the accuracy and decreasing the uncertainty of the estimated growth function, which will further reduce the uncertainty in fishery stock assessment.

**Keywords:** uncertainty; growth function; aging error; tuna

**Key Contribution:** In this paper, the combined effects of aging error and sampled age range on the estimation of tuna body growth were studied, and it is discovered that limited age range was the primary factor leading to biased estimation of growth parameters.

## 1. Introduction

In fishery stock assessment, growth functions are used to provide weight-at-age and length-at-age data for age-structured stock assessment models [1,2]. Various types of growth models have been developed [3,4], and their performance and applicability often vary among different species and populations. In general, the Von Bertalanffy growth function is the most widely applied model in fishery stock assessment, especially among global tuna stocks [5].

The estimation of growth parameters requires accurate measurement of age and length/weight. The measurement of the length and weight is straightforward, while age is often measured by counting deposited growth increments resulted from seasonal changes as the fish grows, such as otoliths, dorsal fin spines, vertebrae, and teeth [3]. However, the age estimated from hard tissues could be inaccurate. External hard tissues of the fish, such as teeth and scales, may be damaged or lost, and the rate of the deposition of growth bands on different hard tissues varies and may fade away. As a result, interpreting increments of different hard tissues from the same individual may give different results due to structural differences. Furthermore, age interpretation may vary among multiple trained personnel,

even for the same hard tissue [6]. In addition, there is a large degree of uncertainty in the aging methodology [1,7], such as daily aging and burning otoliths [8,9]. The age range of sampled individuals is often limited, because it is difficult to obtain sufficient samples that are representative of the whole population. In practice, small- and large-sized individuals are often under-sampled [10], which could lead to a bias in estimating growth parameters.

Tunas are ecologically and economically important species that are widely distributed in tropical, subtropical, and temperate oceans [11]. The sustainable exploitation of global tuna fisheries resources relies on rigorous stock assessment and effective fishery management. Currently, the stock assessment models of major tuna stocks mostly require growth function to obtain the catch-at-age information [12]. Tuna ages are mainly estimated by otoliths, fin spines, vertebrae, and scales, and otoliths can provide more accurate aging results [13–15]. However, the aging errors in otoliths analysis are still significant. Spence and Turtle [16] improved the fit of the Von Bertalanffy growth function by differentiating the spawning season and increasing the amount of data for Atlantic herring (*Clupea harengus*) and Atlantic cod (*Gadus morhua*). Cope and Punt [6] incorporated aging errors in the growth parameter estimation of small fish species by using a random effects framework, but did not consider the effect of age range. Aging error models were developed to improve the estimation of growth parameters of yellowfin tuna (*Thunnus albacares*) [3], but the effect of age range on the estimation of growth curve has not been investigated adequately. Various numerical methods, e.g., bootstrapping, were used to evaluate the uncertainty of estimating growth parameters when aging errors were found [17], but studies regarding estimation uncertainty associated with limited age range are still rare.

In this paper, we examined how the combinations of aging errors and sampled age ranges affected the estimation of Von Bertalanffy growth curves of tuna species. The species considered in this study were southern bluefin tuna (*T. maccoyii*), albacore tuna (*T. alalunga*), yellowfin tuna (*T. albacares*), bigeye tuna (*T. obesus*), and skipjack tuna (*Katsuwonus pelamis*). We first added different levels of aging errors on simulated age and length data based on growth functions from previous tuna studies. Next, we divided the length-at-age data into four groups with different age ranges and fitted the Von Bertalanffy growth model to estimate the growth parameters. Finally, the relative error was used to evaluate the bias in estimated growth parameters.

## 2. Materials and Methods

We used a computer simulation comprising operating models, observation models, estimation models, and evaluation models to study the effects of aging error and age range on estimation accuracy of growth parameters, which is similar to the framework of management strategy evaluation (MSE) [18]. The operating models describe the virtual population dynamics and generate the "truth" [19,20]. The observation models simulate the data-collection process and add errors and uncertainties to the data generated by the operating models [18]. The estimation models use the observed data to estimate the virtual population dynamics. Finally, the evaluation models compare the true and estimated dynamics of the virtual population and quantify the estimation error.

Specifically, our simulation was based on length-at-age data from five tuna species (southern bluefin, albacore, yellowfin, bigeye, and skipjack tuna), i.e., five operating models. Each of the operating models specifies four levels of aging error and four age groups, for a total of 16 observation models. Each observation model was run 1000 times and generated 1000 sets of observed age and length data, and estimation models were fit to each set of data to estimate growth parameters. More details are shown below.

### 2.1. Operating Models

Size and longevity vary between different tuna species, which may affect the results of simulation. To understand the growth pattern of different species and make the simulation more realistic, we configured the operating models based on growth parameters of five tuna species as estimated by previous studies (Table 1).

Albacore tuna reaches a maximum reported age of 14 years [21], an inflection at age 3 [22], and an asymptotic length at age 7 or 8 [4].

The maximum reported age of bigeye tuna was 16 years [4], an inflection occurred at age 2~3 [23,24], and asymptotic sizes were attained at 8 years [13].

The longevity of yellowfin tuna can be 18 years [25]. The first phase of growth is slow until the fork length reaches 65~75 cm [24,26,27]. In the second phase, the growth rate reaches a peak, and then decreases when individuals grow to ~145 cm. Eveson et al. [24] found that the inflection point was between age 2 and 3.

Southern bluefin tuna is a long-lived species of approximately 41 years old [7]. Growth is relatively rapid in the first few years of life, and asymptotic length is reached at 20 years [4]. Hearn et al. [28] found an inflection at the fork length of 85 cm. Polacheck et al. [29] suggested that the growth rate of the first few years increased with time.

Skipjack tuna is the fastest-growing species among all tuna species, and the maximum age is about 6~7 years [4]. The growth is rapid until the fork length reaches 40~50 cm.

**Table 1.** Growth parameters of five tuna species.

| Species [1] | $L_\infty$ (cm) | K (Year [1]) | $t_0$ (Year [1]) | Inflection Points (Year) | Longevity (Year) |
|---|---|---|---|---|---|
| Alb | 104.52 | 0.4 | −0.49 | 3 and 7~8 | 14 |
| BET | 151.1 | 0.386 | −0.410 | 2~3 and 8 | 16 |
| YFT | 153.3 | 0.36 | −0.8 | 2~3 and 7~8 | 18 |
| SBT | 183.18 | 0.185 | −0.923 | 3 and 20 | 41 |
| SKJ | 122.5 | 0.12 | −1.69 | 1 | 6~7 |

[1] ALB: albacore; BET: bigeye; YFT: yellowfin; SBT: southern bluefin tuna; SKJ: skipjack.

The dataset of length-at-age was simulated according to the Von Bertalanffy growth model:

$$L_t = L_\infty \times \left(1 - e^{-k \times (t-t_0)}\right) \tag{1}$$

where $L_t$ is the length at age in cm, $L_\infty$ is the asymptotic length in cm, k is the growth coefficient in year$^{-1}$, and $t_0$ is the theoretical age in the year at zero length.

## 2.2. Observation Models

Based on the longevity of five tuna species, we simulated four ranges of sampled age for each species (Table 2). Among tuna species, only skipjack tuna has one growth inflection point and is divided into three age groups due to its short life span.

The intervals between age groups were generally 1 year; however, yellowfin tuna and bigeye tuna were set to half a year in the Young group, and skipjack tuna was set to a quarter of a year in the Young group.

**Table 2.** Sampled age ranges of five tuna species.

| Specie | Young Group | Intermediate Group | Old Group | Full-Age Group |
|---|---|---|---|---|
| ALB | 1~3 | 4~7 | 8~14 | 1~14 |
| BET | <2.5 | 3~7 | 8~16 | 0.5~16 |
| YFT | <1.5 | 2~7 | 8~18 | 0.5~18 |
| SBT | 1~3 | 4~20 | 21~41 | 1~41 |
| SKJ | <1 | – | 2~7 | 0.25~7 |

To investigate the effect of aging error on parameter estimation, we assumed that the observation errors were mainly from the process of aging rather than measuring length. Therefore, we fixed the measurement error of length data at a low level (e.g., $\sigma_L = 0.01$), and specified four scenarios of aging error in the simulation (Table 3). The aging errors were independent and normally distributed around zero with a constant standard deviation

(SD) in scenarios 1, 2, and 3, i.e., $\sigma_t = 0.01$, 0.1, 0.25 in S1, S2, and S3. In scenario 4, the aging error was set to increase with age, i.e., $\sigma_t$ increases from 0.1 to 0.25 over the entire age range. The formulas are as follows:

$$t_e = t \times e^{\varepsilon_t}; \; \varepsilon_t \sim N(0, \; \sigma_t) \tag{2}$$

$$L_e = L_t \times e^{\varepsilon_L}; \; \varepsilon_L \sim N(0, \sigma_L) \tag{3}$$

**Table 3.** Error settings of different observation models.

| Scenario | Error | Age Groups |
|:---:|:---:|:---:|
| S1 | $\varepsilon_t \sim N\,(0, \; 0.01)$ | Young group |
| | $\varepsilon_t \sim N\,(0, \; 0.01)$ | Intermediate group |
| | $\varepsilon_t \sim N\,(0, \; 0.01)$ | Old group |
| | $\varepsilon_t \sim N\,(0, \; 0.01)$ | Full-age group |
| S2 | $\varepsilon_t \sim N\,(0, \; 0.01)$ | Young group |
| | $\varepsilon_t \sim N\,(0, \; 0.01)$ | Intermediate group |
| | $\varepsilon_t \sim N\,(0, \; 0.01)$ | Old group |
| | $\varepsilon_t \sim N\,(0, \; 0.01)$ | Full-age group |
| S3 | $\varepsilon_t \sim N\,(0, \; 0.25)$ | Young group |
| | $\varepsilon_t \sim N\,(0, \; 0.25)$ | Intermediate group |
| | $\varepsilon_t \sim N\,(0, \; 0.25)$ | Old group |
| | $\varepsilon_t \sim N\,(0, \; 0.25)$ | Full-age group |
| S4 | $\varepsilon_t \sim N\left(0, \; 0.1 + \frac{0.15}{n} \times (i-1)\right)$ | Young group |
| | $\varepsilon_t \sim N\left(0, \; 0.1 + \frac{0.15}{n} \times (i-1)\right)$ | Intermediate group |
| | $\varepsilon_t \sim N\left(0, \; 0.1 + \frac{0.15}{n} \times (i-1)\right)$ | Old group |
| | $\varepsilon_t \sim N\left(0, \; 0.1 + \frac{0.15}{n} \times (i-1)\right)$ | Full-age group |

n: age interval; i: ith age.

### 2.3. Estimation Models

For each observation model, we generated 1000 sets of length-at-age data, which was fit to the Von Bertalanffy growth model. Maximum likelihood estimation was used to estimate parameters. The negative log-likelihood function is

$$-\log(L) = -\sum_{i=1}^{n} \ln\left(\frac{1}{\sqrt{2\pi}\sigma} e^{-\frac{(L_i - \hat{L}_i)^2}{2\sigma^2}}\right) \tag{4}$$

where $L_i$ is the observed length at age i, $\hat{L}_i$ is the estimated length at age i, and n is the number of ages.

### 2.4. Evaluation Models

The relative error $\delta$ was used to evaluate the bias between the estimated growth parameters $\hat{A}$ from the estimation models and true growth parameters A from the operating models. $\delta$ of each set in one dataset was calculated as follows:

$$\delta = \frac{\hat{A} - A}{A} \times 100\% \tag{5}$$

All the procedures were conducted in R (reversion 4.1.3), and the package "stats4" was used to estimate the growth parameters.

## 3. Results

### 3.1. Estimation Uncertainty of the Growth Curve

The estimated growth curves had similar patterns among five tuna species. The distribution of the growth curve of albacore tuna is shown in Figure 1, and the distributions of bigeye, southern bluefin, yellowfin, and skipjack tuna are shown in Figures S1–S4. In

scenarios 1, 2, and 3, the estimated growth curves became more uncertain when aging error increased. When SD increased over age (scenario 4), the curves were well estimated when all age groups were sampled, but became uncertain when only part of the age groups were sampled.

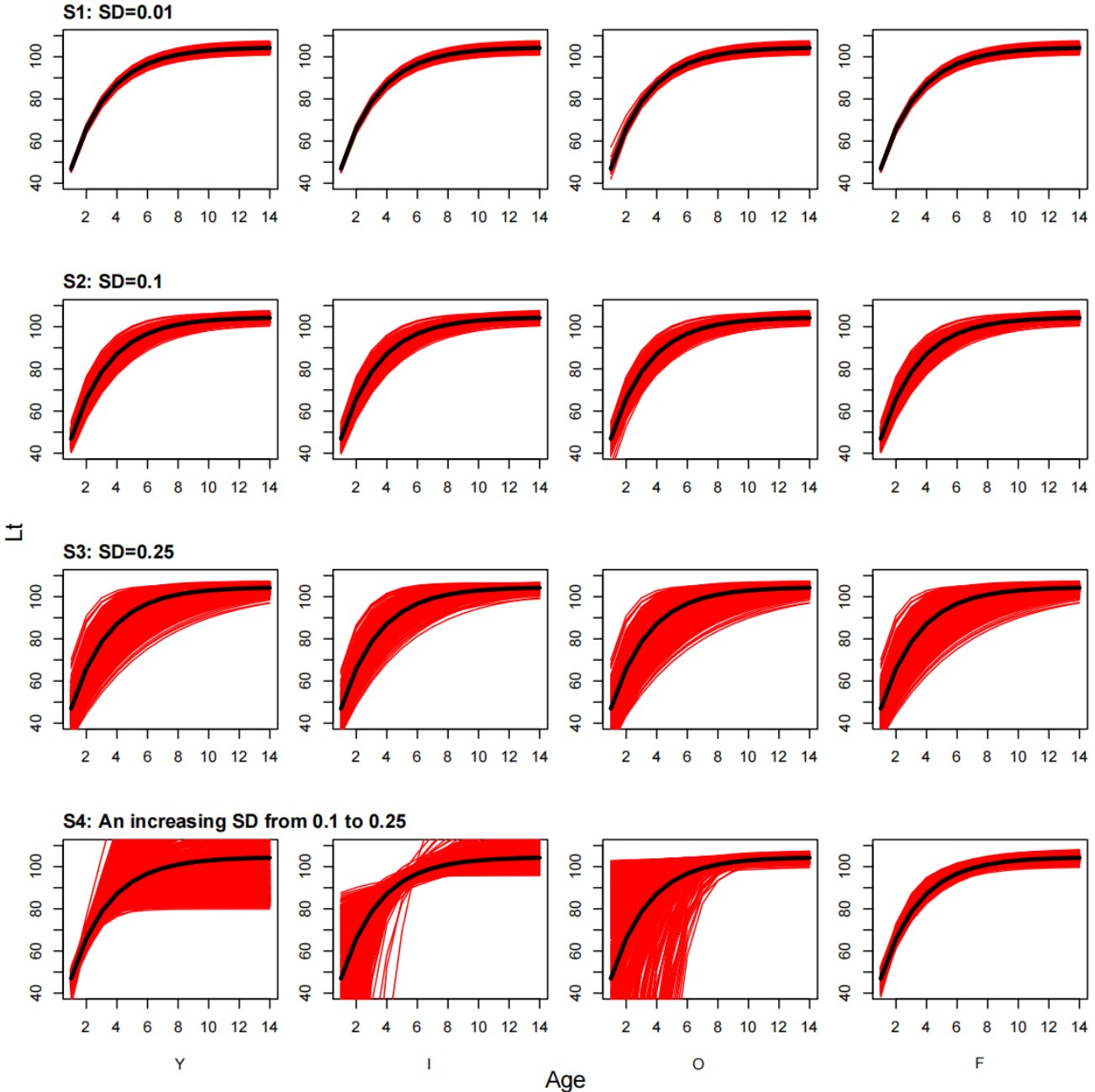

**Figure 1.** Distributions of estimated growth curve of simulated albacore tuna (the red curves are the simulated curves, and the black curve is the initial growth curve; Y: Young group, I: intermediate group, O: Old group, F: Full-age group, the same below).

*3.2. Parameter Distributions*

When the aging error was fixed between ages (S1–S3), the estimation errors of three growth parameters were similar among different age ranges for albacore and bigeye tunas (Figures 2 and 3). For southern bluefin and yellowfin tunas, the parameter $L_\infty$ was underestimated when only young age groups were sampled (Figures 4 and 5). For skipjack tuna, the parameter $L_\infty$ was underestimated and the parameter k was overestimated when

only young age groups were sampled (Figure 6). The estimation uncertainty of three parameters was similar among different age ranges for all species.

When aging error increased with age (S4), the estimation errors of three growth parameters were greater when not all age groups were sampled. The estimation uncertainty of parameters $L_\infty$ and k was greatest when only young age groups were sampled, while the estimation uncertainty of parameters $t_0$ was the greatest when only old age groups were sampled. These patterns of estimation errors and estimation uncertainty were consistent over all five tuna species (Figures 2–5 and S5–S19).

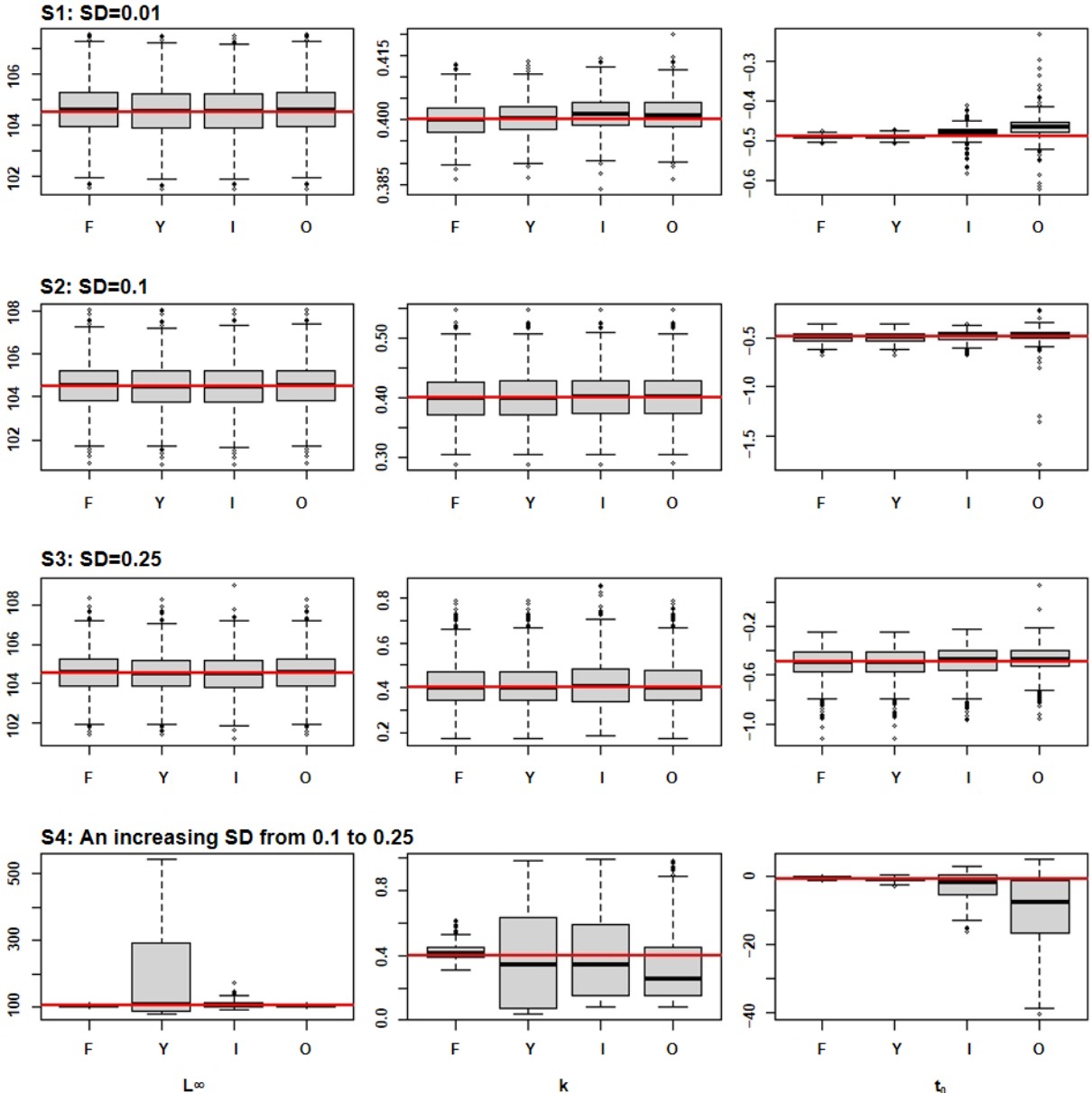

**Figure 2.** Distributions of estimated growth parameters of simulated albacore tuna (the red line is the initial parameter; $L_\infty$, k and $t_0$ are three parameters in the Von Bertalanffy growth model).

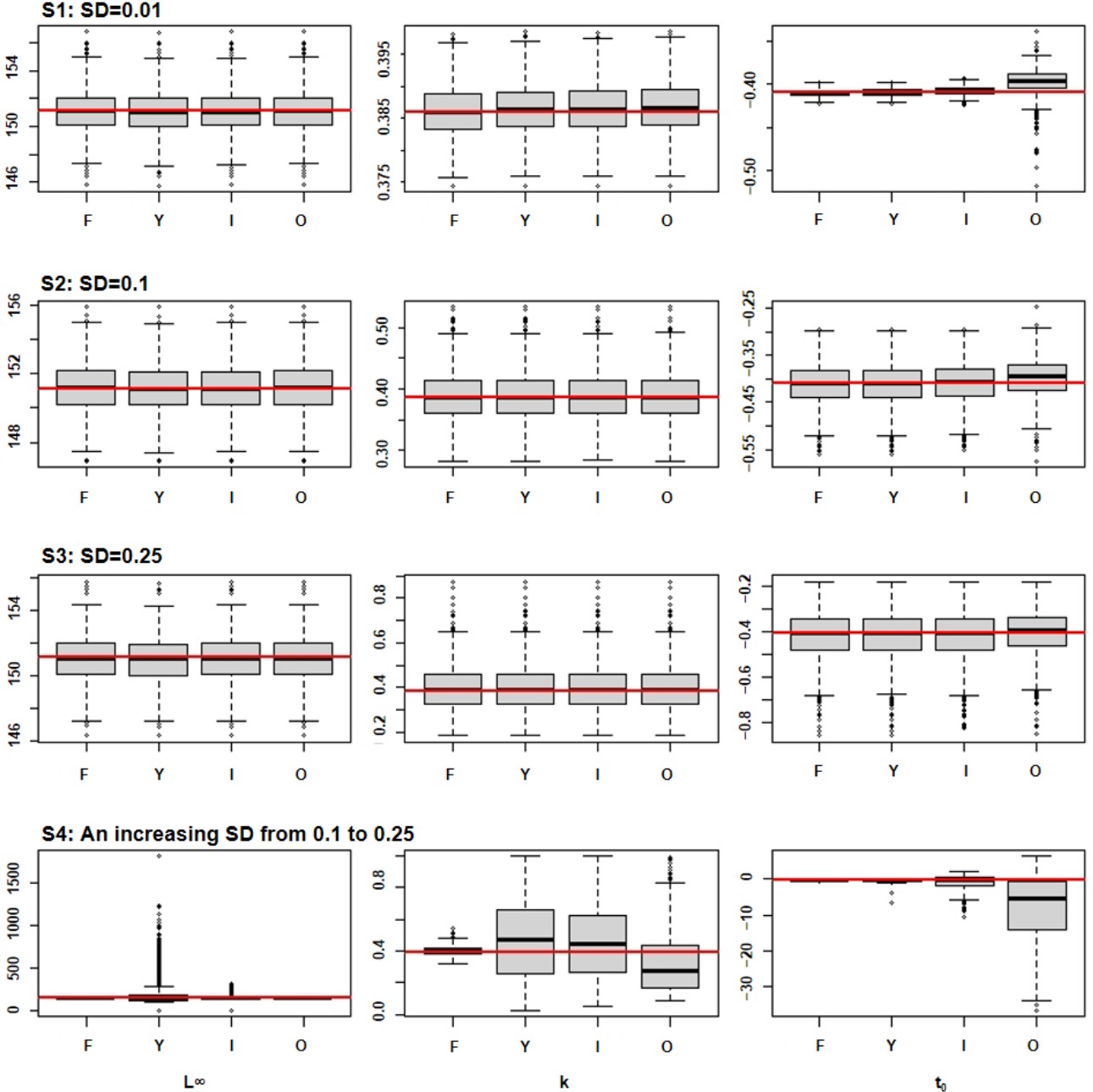

**Figure 3.** Distributions of estimated growth parameters of simulated bigeye tuna (the red line is the initial parameter; $L_\infty$, k and $t_0$ are three parameters in the Von Bertalanffy growth model).

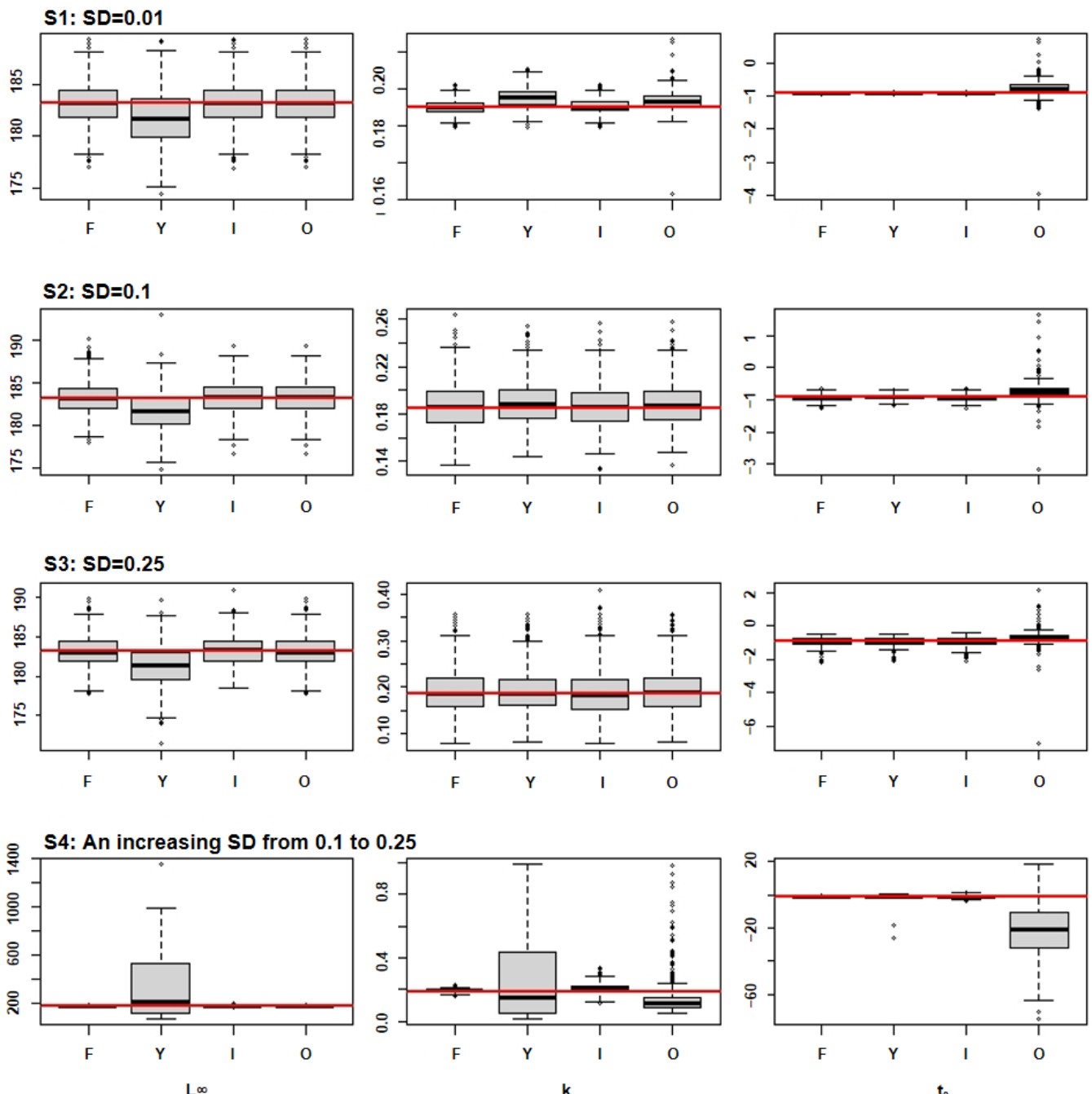

**Figure 4.** Distributions of estimated growth parameters of simulated southern bluefin tuna (the red line is the initial parameter; $L_\infty$, k and $t_0$ are three parameters in the Von Bertalanffy growth model).

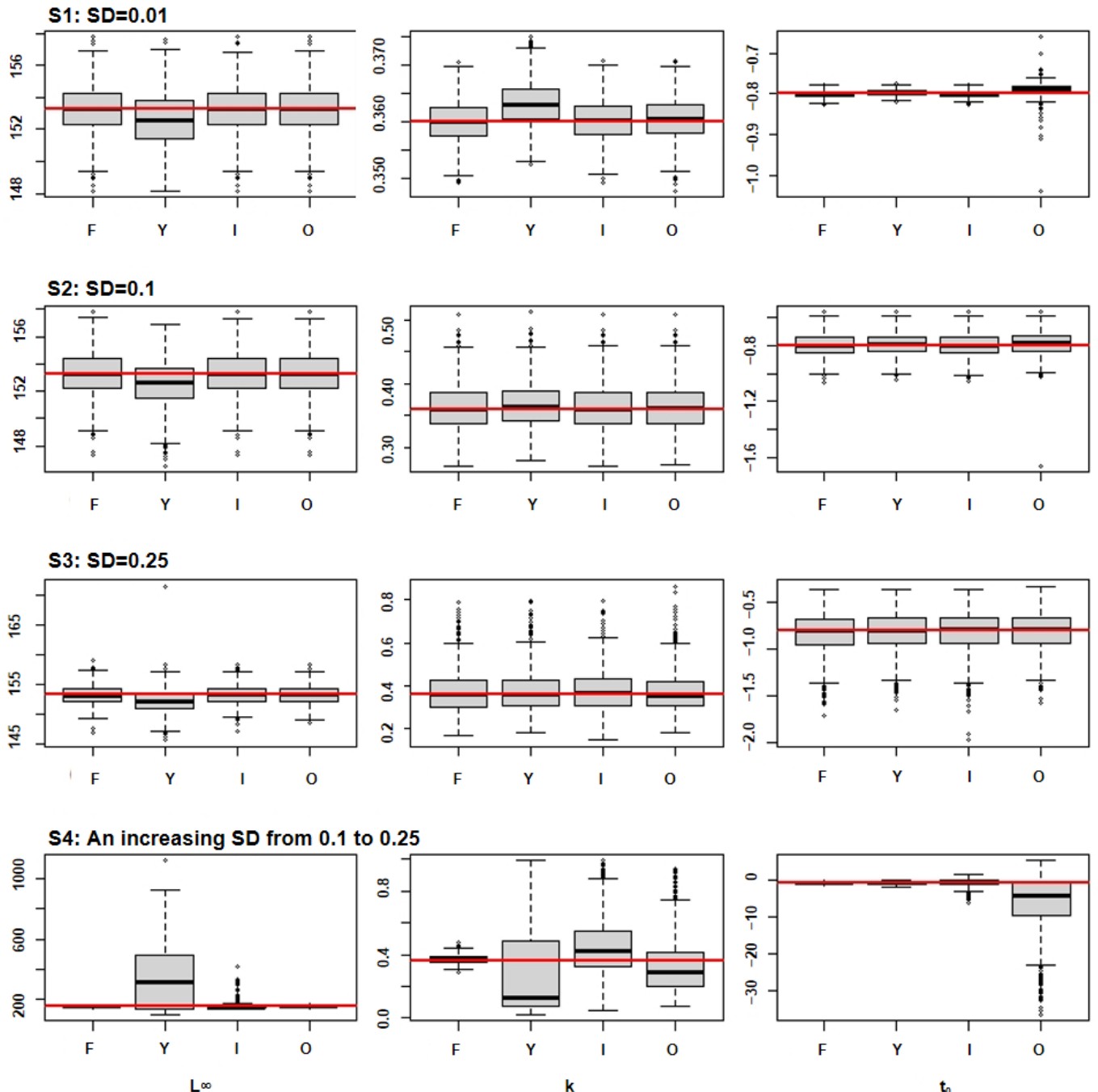

**Figure 5.** Distributions of estimated growth parameters of simulated yellowfin tuna (the red line is the initial parameter; $L_\infty$, k and $t_0$ are three parameters in the Von Bertalanffy growth model).

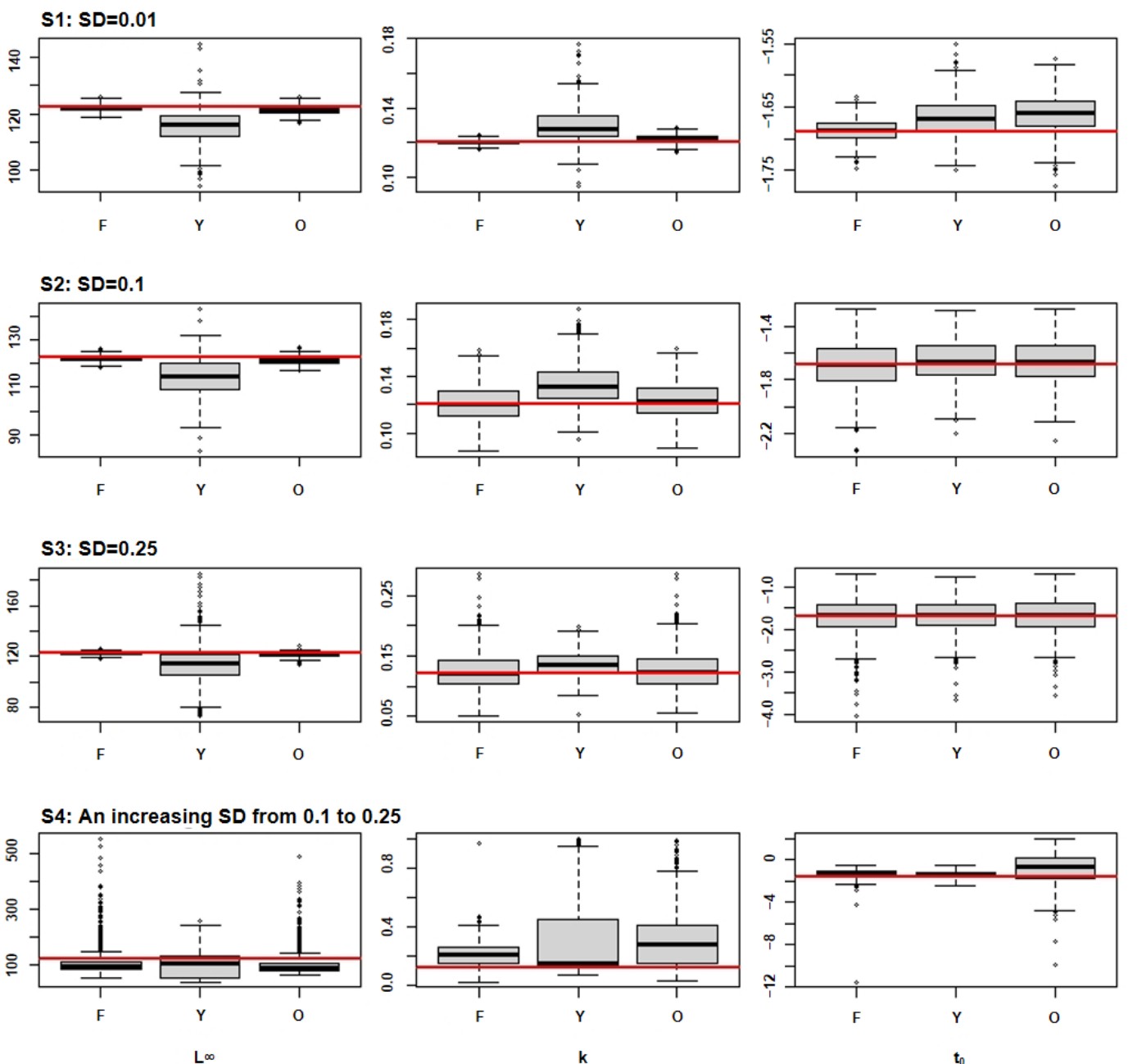

**Figure 6.** Distributions of estimated growth parameters of simulated skipjack tuna.

## 4. Discussion

The growth parameters estimated for the same species, or even population, could be different among studies [30,31], and these differences were evident in the majority of tuna growth studies [32–34], which may be caused by changing fish-growth patterns due to environmental and anthropogenic effects (e.g., climate change, fishing, ecosystem change, etc.). For example, warming temperature caused by climate change may increase the growth rate, while overfishing of prey species could decrease the growth rate of predatory fish [17,34,35]. Alternatively, aging uncertainty could also lead to different results of growth patterns among studies, even when the actual growth pattern remains constant. However, studies on the latter were relatively few. In this study, we implemented a simulation analysis to study the effects of aging error and sampled age range on the estimation of growth curves of major tuna species [36].

Aging error is inevitable, regardless of the methods used for aging. Analysis of length–frequency data, tagging–recapture studies, and interpretation of hard tissues are major methods for interpreting age in tuna species [25]. Growth increments formed on hard tissues can provide more accurate age interpretation than length–frequency and tagging–recapture studies [37], and this method also has the advantage of convenient sampling. However, differences between hard tissues can affect the difficulty and accuracy of age interpretation. The area of vascularization increases with age, making it difficult to interpret growth bands on dorsal fin spines, but vertebrae can accurately estimate the age of the first few years [7,25]. Because of the presence of more defined growth bands on the otoliths, age interpretation can be more accurate than other hard tissues [38]. Farley et al. [13] calculated the relationship between otolith size and age, which yielded more accurate decimal age. The process of preparing and interpreting increments depends on proficiency, and discrepancies between repeated results may increase with the age of tuna [34,36]. As a result, different trained personnel and institutions can produce contradictory age results, especially for tuna distributed in tropical or subtropical areas with limited seasonal variation [23,39]. Although the average percent error (APE) and the coefficient of variation (CV) are frequently used to reduce the bias of trained personnel [10,36,40], or a confidence score is assigned to each structure [7,10], no standard exists to determine the accuracy of age identification due to the subjective judgment of the trained personnel [41]. To reduce the influence of aging errors, many studies often contained more samples and expanded the age range [39,42]. Establishing a consistent and standardized process, as well as additional training, can aid in improving aging accuracy and comparing different studies [36]. The re-estimation of previous samples with validated information on the formation of increments may update the results [43].

We simulated the effects of different aging errors on estimation of growth curves. We discovered that as the aging error increased, so did the uncertainty of estimated growth parameters. Because errors in the interpreting process can affect the estimation of parameters, uncertainty must be taken into account in age interpretation [6]. However, the accuracy and level of uncertainty of identification vary due to the different techniques or hard tissues in some traditional growth studies, which is why there was a large bias among previous studies [8,44,45]. Pacicco et al. [25] used otoliths to estimate the maximum age of yellowfin tuna, which was three times higher than the results of Lessa and Duarte-Neto [46] using spines, though the length range was similar. Stequert et al. [8] and Schaefer et al. [45] separately aged yellowfin and bigeye tuna using daily increments in otoliths, but underestimated the age, resulting in a larger $L_\infty$ and low k value. In addition, aging errors tend to increase with age [26], adding major uncertainty to the growth estimation. Considering the uncertainty of aging is important for growth estimation, just as the aging error model developed by Dortel et al. [3] can estimate age more accurately than traditional methods. Cope and Punt [6] also found that incorporating uncertainty in estimating growth was more accurate.

Previous studies revealed that the growth rate of different phases varies widely and has a high degree of error [34], and a limited age range in the fitting process may lead to bias in growth estimation. Due to gear selectivity and minimum size limitations, small fish are often missing in the catch [4]. Especially for tuna, a pelagic migratory species, fishing may only occur in specific areas [4,35]. In addition, large individuals are also difficult to catch due to their lower abundance in fish populations [47]. Therefore, hard tissues can only be obtained from the fish within a limited body length range, and data sources in studies may not cover all groups of age [25]. Back-calculation has been used in many studies to estimate the growth of small individuals, but there are large differences between the parameters estimated in this way and those estimated directly by hard tissues [40,46]. It is also necessary to consider whether the acquisition of hard tissues will affect the economic value of the fish during the sampling process. For instance, otolith extraction interferes with the market value [48], making it difficult to obtain sufficient representative numbers of otolith samples. In future studies, samples from different types of fishing gear or joint

research with countries or organizations can be more representative and improve the estimated parameters [41,49].

In our study, we found that when accuracy is ensured, parameters may be unaffected by age range, but it is an idealized state; i.e., the increments on hard tissues are clearly rendered and accurately interpreted by different trained personnel. Obtaining such a growth estimation is particularly challenging, since it involves not only a highly defined set of interpretation criteria, but also a high level of expertise gained via extensive training. We found that parameter $L_\infty$ and k of skipjack tuna had greater estimation errors when only young groups were sampled (S1–S3), regardless of the levels of aging error, which is probably related to their growth rate, as skipjack tuna is known to be the "fastest growing" of tuna species [4,50]. There are numerous growth studies of skipjack tuna with large variation of growth parameters, even with the use of otolith daily aging [51]. When simulating the growth of skipjack tuna, much higher $L_\infty$ and lower k were used, and the asymptotic length was greater than the maximum size in samples. Despite broadening the age range, no obvious second inflection was observed. We argue that changes in age ranges may have a more significant impact on fast-growing species. When the aging error was fixed, the parameter k of southern bluefin tuna had a greater estimation error when only old groups were sampled. In old groups, the growth is slow or even stopped [4], increasing the difficulty of estimating the age–length relationship. The effect of age ranges on growth was well-represented in S4, where estimation error and uncertainty increased when only part of age groups were sampled. The growth of bigeye tuna, estimated using a 1~9 year age range [40], was compared to the age range of 1~16 years [52], with the latter yielding $L_\infty$ close to the actual sample length. As noted by Neilson and Campana [34], the difference in results between Turner and Restrepo [53] was most likely related to the differences in size and age ranges.

Many growth studies attempt to increase sample size and include as many age groups as possible, so that more representative length-at-age data can be obtained. Using these data for growth estimation can reduce the estimation uncertainty [17]. When the sample size is small, the uncertainty of parameter estimation increases, and using the mean length of each age can appropriately reduce the effect of observation error. There is some variation in the parameters estimated by the dataset and average length-at-age, but with improved fishery management and access to data, the large sample approach becomes more appropriate for studying the growth of tuna. Alternatively, the bootstrapping method and the Markov Chain Monte Carlo method have been used to quantify the uncertainty of the parameters by repeated sampling [54]. The simulation analysis in this paper is another approach to quantify estimation uncertainty and errors, which can be used to test the sensitivity of results to different types of aging uncertainties. Although numerous methods exist, no single method can fully account for aging uncertainty in growth studies. Therefore, a combination of multiple methods is needed to increase the accuracy in estimating growth curves.

Growth information is critical for conducting accurate stock assessments [12] and can be used to construct age compositions of populations to improve the results of stock assessments [17]. Age structure and individual size can be used to assess the population status, and the change of increments in hard tissues can be used to analyze whether the population is affected by environmental changes [12,35]. According to Juan-Jorda et al. [55], growth rates and longevity were the best predictors of changes in population resources. However, there is uncertainty in estimating growth by hard tissues, and the age structure may be truncated as fishing intensity increases [47], resulting in large changes in the estimates of these parameters. Fishery stock assessment results are sensitive to changes in growth parameters [10], which have a direct impact on estimates of fishing mortality and reproduction, and are ultimately used to infer stock status and biological reference points [25,38,56]. Along with the input of growth parameters, uncertainties in the process of age identification can be transferred to the resource assessment, making the estimates of the total allowable catch (TAC) too optimistic for short-term management of the stock [57]. Williams et al. [58] believed that the growth of the South Pacific albacore is subject to

longitudinal and sex-dependent variations that cannot be estimated by a single growth curve or used in the stock assessment. Wells et al. [10] estimated the spawning stock biomass of North Pacific albacore using two sets of parameters and discovered a several-fold difference in results. Thus, a combination of different age identification methods, such as tag–recapture and length–frequency data is the most direct and effective way to verify growth and reduce the influence of inherent uncertainties from hard tissues. The majority of the world's tuna stocks are fully or over-exploited, but the human demand for tuna continues to grow [4]. We need to maintain the balance between economic development and the sustainable exploitation of fish stocks, particularly for a tuna species with high economic value (e.g., bluefin tuna and bigeye tuna). Therefore, in addition to obtaining more accurate growth estimation for effective stock assessment [47], regional fisheries management organizations (RFMOs) should pay more attention to protecting age structure, which is important for maintaining the productivity and stability of populations [4].

## 5. Conclusions

Simulation was used to examine the effects of aging error and age range on the estimation of growth parameters in five tuna species. Both factors influenced the uncertainty of growth estimation, with the sampling age range having a greater impact on growth parameter estimation. As a result, future studies should focus on improving the accuracy of age interpretation and expanding the sampled age range to reduce uncertainty in the process of growth estimation, which will also improve assessment accuracy.

**Supplementary Materials:** The following supporting information can be downloaded at https://www.mdpi.com/article/10.3390/fishes8030131/s1: Figure S1. Distributions of estimated growth curve of simulated bigeye tuna; Figure S2. Distributions of estimated growth curve of simulated southern bluefin tuna; Figure S3. Distributions of estimated growth curve of simulated yellowfin tuna; Figure S4. Distributions of estimated growth curve of simulated skipjack tuna; Figure S5. Distributions of estimated relative error of $L_\infty$ of albacore tuna; Figure S6. Distributions of estimated relative error of k of albacore tuna; Figure S7. Distributions of estimated relative error of t0 of albacore tuna; Figure S8. Distributions of estimated relative error of $L_\infty$ of bigeye tuna; Figure S9. Distributions of estimated relative error of k of bigeye tuna; Figure S10. Distributions of estimated relative error of t0 of bigeye tuna; Figure S11. Distributions of estimated relative error of $L_\infty$ of southern bluefin tuna; Figure S12. Distributions of estimated relative error of k of southern bluefin tuna; Figure S13. Distributions of estimated relative error of t0 of southern bluefin tuna; Figure S14. Distributions of estimated relative error of $L_\infty$ of yellowfin tuna; Figure S15. Distributions of estimated relative error of k of yellowfin tuna; Figure S16. Distributions of estimated relative error of t0 of yellowfin tuna; Figure S17. Distributions of estimated relative error of $L_\infty$ of skipjack tuna; Figure S18. Distributions of estimated relative error of k of skipjack tuna; Figure S19. Distributions of estimated relative error of t0 of skipjack tuna; Figure S20. A fitted curve (red line) and actual curve of length-at-age (black line) (black dots are the noise data); Figure S21. Standard deviation estimated from each model of albacore; Figure S22. Standard deviation estimated from each model of bigeye tuna; Figure S23. Standard deviation estimated from each model of southern bluefin tuna; Figure S24. Standard deviation estimated from each model of yellowfin tuna; Figure S25. Standard deviation estimated from each model of skipjack tuna.

**Author Contributions:** Conceptualization, D.L. and F.Z.; methodology, D.L.; validation, D.L.; formal analysis, D.L.; resources, F.Z.; writing—original draft preparation, D.L.; writing—review and editing, F.Z. and Q.L.; visualization, D.L.; supervision, J.Z.; funding acquisition, F.Z. All authors have read and agreed to the published version of the manuscript.

**Funding:** This research was funded by the National Natural Science Foundation of China (32002393); Project on the Survey and Monitor-Evaluation of Global Fishery Resources sponsored by the Ministry of Agriculture and Rural Affairs (2021-0109).

**Institutional Review Board Statement:** The study was approved by the Institutional Animal Care and Use Committee of Shanghai Ocean University (protocol code 2022031501, approved on 15 March 2022).

**Informed Consent Statement:** Not applicable.

**Data Availability Statement:** Data is contained within the article.

**Acknowledgments:** We thank those who provided help with the code. We thank the Program on the Survey, Monitoring and Assessment of Global Fishery Resources sponsored by the Ministry of Agriculture and Rural Affairs of China.

**Conflicts of Interest:** The authors declare no conflict of interest.

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
