# Peer review of "Effects of Aging Uncertainty on the Estimation of Growth Functions of Major Tuna Species"

_fishes, doi:10.3390/fishes8030131_

Round 1
Reviewer 1 Report
In this manuscript the authors describe a modeling approach to determining the potential impact of aging error/uncertainty on growth parameters of five different tuna species. The manuscript provides information about the potential impacts of misinterpreting tuna ages. Though others have looked at some of these, or other species of tuna, this set of species has not been evaluate in this way before. Although the information that the manuscript relies on is useful, the framing and presentation of the information needs to be improved to increase the value of this contribution.
I have made detailed comments in the Introduction, but similar issues remain throughout the manuscript.
Comments on the Introduction:
Lines 33-36 are awkwardly worded/presented. The topic sentence is relatively uninformative, and the first four sentences of this paragraph essentially say that interpreting fish age with hard structures can be challenging, but is needed to characterize how they grow.
Line 36: the term “on the other hand” could be replaced with “alternatively”.
Line 37: The term interpret should be used versus “reading” hard structures for aging.
Line 38: Both sentences begin with “In addition” which detracts from the verbiage.
Line 46: “Nowadays” should be replaced with “now”, or “currently”.
Line 48: The text mentions that recently (referring to a 2012 manuscript) otoliths were shown to be a reliable aging structure in large pelagic tuna. There are more recent works with more tuna species than what is referenced here. Also, this is a relatively unimportant point as otoliths have long been recognized as be the gold standard for most fish species for age interpretation.
Line 50: This sentence structure is incorrect and needs to be rewritten.
Line 60: This sentence is redundant with the sentence that follows in the methods section stating, “Meanwhile, the relative error is used to examine the bias of various parameters”. If that sentence is retained, the word “meanwhile” should be removed.
Other issues throughout the manuscript:
In the Operating Models section in the Methods, the information in paragraphs 2-6 could be shown in a small table.
Throughout the Methods section there is verbiage that might be more appropriate for the Introduction or Results sections.
Are Low, Intermediate, and High the correct terms to use for age? Consider Young, Intermediate, and Old.
I don’t understand or can’t find what is being referred to when Figures 2-5; S5, S8, S11, and S14 are mentioned. I was unable to access “S1” mentioned in the Supplementary Materials, but it doesn’t seem like that information would be there based on the description provided. What are S5, S8, S11, and S14, and do S6, S7, S9, S10, S12, S13, or others exist?
The terms success, converge/convergence/convergent/convert need to be dealt with more clearly throughout the Results.
The Discussion section relies heavily on the conclusions of others (e.g., Murau e tal. 2017) and needs to expand upon the implications from the work presented here specifically. The study looked at aging uncertainty and how that might contribute to differences in growth parameters in tuna over time. However, there is not verbiage about how these findings could contribute to explaining (or not) these differences. The conclusion largely describes the need for more accurate age interpretation, but the biases or management errors that could occur from not doing so are only alluded to without detail.
Reviewer 2 Report
The article presents a statistical study of uncertainty in the aging of different types of fish. The uncertainty in the parameters of the model is quantified using maximum likelihood estimation on different synthetic data sets of length vs age. The article presents a good case for publication and the authors are recommended to address the below issues before being considered for publication.
1) Please do thorough English and grammar checks.
2) in line 96, it is mentioned that "The dataset of length-at-age was simulated according to the Von Bertalanffy growth 96 curves, and the initial parameters were shown in Table 1". How are the initial parameters assumed? Did you fit the real data with the Von Bertalanffy equation and obtained these parameters?
3) The study mainly revolves around operating models, observation models, and estimation models. For completeness, please add an explanation of these models and how they are different from each other with references.
4) In Figures 1-6, the standard deviation was mentioned as S1, S2, etc It is confusing, please mention SD=value of S1 instead for clarity.
5) It is mentioned in the paper that 1000 sets of data are produced by the observation model. If I understand correctly, each set of data is length vs age and gaussian error is added to the data. In such case, the data should be noisy but the curves in Fig.1 show very smooth. Please plot a single set of the curve with noise added along with the actual curve of length vs age to show the noise. Please plot noisy data as a scatter plot.
6) This article talks about quantifying uncertainty in estimated parameters by fitting the model to 1000 sets of length vs age data with added noise and reporting the statistics of the estimated parameter. For each parameter estimate or model, one data set is used and a total of 1000 values of parameters of estimated. In a real case, the data will be available as a single set and even if the large set of data is divided into smaller sets, the data set will not follow the observational model with a particular parameter and there will be huge errors in the fit resulting in a very wide parameter estimation range. Did the authors consider other uncertainty quantification techniques in parameters such as Markov chain monte Carlo methods and quantify uncertainty in parameters by considering the entire dataset in a single go? Please add a discussion of how the method presented in this article is different from quantifying uncertainty in parameters in a single go.
7) The introduction is short and references to uncertainty quantification methods are lacking.
8) Along with the parameter distribution, plot the box plot of standard deviation estimated from each model fit and data in figures 2 -5.
9) The entire article is based on fitting the model to the synthetic data generated by the observation model. It adds more value to the article if the authors present a case study with real data from the field.
Round 2
Reviewer 1 Report
Many of my comments were addressed. and the writing and overall presentation has been improved. However, there are still locations in the manuscript that could benefit from more attention beyond the comments I provided initially. The presentation of the material and text throughout could be improved.
Reviewer 2 Report
Thank you for the response. I still see the standard deviation on the title of the figures as SD1 but not SD1=value. Please modify the figures. Additionally, in response 5, the synthetic data are not noisy. If a normally distributed error is added to the true value, the synthetic data looks as shown in this article fatigue damage diagnostics–prognostics framework for remaining life estimation in adhesive joints. But in this case, the synthetic data is smooth and not noisy. Additionally, the response to point 6 is not appropriate and I don't see any work related to parameter estimation using the Markov chain Monte Carlo method is not discussed. Please modify the figures and the response to 6 approprtiately.
Author Response
Response: Thanks again for your comments, in the first revision we put the changes to SD in the caption under the picture, here we made the changes as you requested. We have adjusted the last paragraph to 343 lines. In line 344, “as a way to obtain more representative length-at-age data” is changed to “, so that more representative length-at-age data can be obtained”.in line 346, “this method increases the uncertainty of estimating parameters” is changed to “the uncertainty of parameter estimation increases”. “Markov Chain Monte Carlo method” is inserted in line 348, and the word “determine” is changed to “quantify”. we delete the last sentence in this paragraph, and add “The simulation analysis in this paper is another approach to quantify estimation uncertainty and errors, which can be used to test the sensitivity of results to different types of aging uncertainties. Although numerous methods exist, no single method can fully account for aging uncertainty in growth studies. Therefore, a combination of multiple methods is needed to increase the accuracy in estimating growth curves.”
Round 3
Reviewer 1 Report
Some adjustments in the manuscript have been made, but changes remain that should be made throughout the manuscript, not only the places where comments were made directly. For example, the use of the term "reading" does not really apply to otoliths or fish age interpretation using hard structures. The term "read" has been replaced widely by the word "interpret". There is nothing written on otoliths or fin rays or scales, so it is generally agreed that trained personnel interpret fish ages from these structures, rather than "readers" reading ages on an otolith or other aging structure. When the comment was made to use the word "interpret" rather than "read", that comment applies throughout the entire manuscript, not just where this comment was made directly. Instead, this term is now used more often in newly added text, and was not addressed when used extensively in the Discussion. Similarly, other comments made that were pointed out in the Introduction directly also extent to the Methods, and especially the Discussion section. The added context and framing in the Discussion was helpful, but can be polished further.
Reviewer 2 Report
Thank you for modifying the article as per the recommendation
Author Response
Dear Reviewer:
Thank you for the first two improvements to our manuscript (ID: fishes-2196089). These comments are valuable and helpful to us in revising and improving the paper, as well as providing important guidance for our research. If there are still problems, we will correct them carefully and would greatly appreciate your further comments.